# Risk Factors Including Age, Stage and Anatomic Location that Impact the Outcomes of Patients with Synovial Sarcoma

**DOI:** 10.3390/medsci6010021

**Published:** 2018-03-06

**Authors:** Minggui Pan, Maqdooda Merchant

**Affiliations:** 1Department of Oncology and Hematology, Kaiser Permanente, 710 Lawrence Expressway, Santa Clara, CA 95051, USA; 2Kaiser Permanente Division of Research, Oakland, CA 94612, USA; maqdooda.merchant@kp.org

**Keywords:** synovial sarcoma, soft tissue sarcoma, chemotherapy, overall survival, disease-free survival, metastatectomy

## Abstract

Synovial sarcoma is a high-grade soft tissue sarcoma that inflicts mostly children and young adults with high mortality rate; however, the risk factors that impact the outcomes remain incompletely understood. We have identified the synovial sarcoma cases from the Kaiser Permanente Northern California cancer registry between 1981 and 2014. Kaplan–Meier plots were used to display disease-free survival (DFS) and overall survival (OS); log-rank tests and Cox proportional hazard models were used to determine the impact of clinical factors on DFS, OS, and disease-specific survival. Tumor size > 5.0 cm and age > 50 years were associated with higher risk of presenting with stage IV disease. Median OS for patients with stage IV was 1.3 years and 7.8 years for early-stage disease. For patients with early-stage disease, tumor size > 5.0 cm was significantly associated with worse DFS, sarcoma-specific morality, and OS. Compared to extremity primary, patients with head and neck and trunk primary had approximately three-fold higher sarcoma-specific mortality and lower OS. There was no significant difference in DFS or OS among three histologic subtypes. Pre-operative and/or post-operative chemotherapy was not associated with improvement in DFS or OS. Twenty-six patients relapsed with predominantly lung metastasis, thirteen of whom received metastatectomy with a median OS of 7.8 years, compared to 2.3 years for patients who did not receive metastatectomy. In conclusion, age older than 50 years and tumor size > 5.0 cm are risk factors for presenting with stage IV disease. For early-stage patients, trunk and head and neck primary as well as tumor size > 5.0 cm are risk factors for decreased OS.

## 1. Introduction

Synovial sarcoma is a high-grade soft tissue sarcoma (STS) with three histological subtypes (biphasic, monophasic, and poorly differentiated) that are characterized by variable degree of spindle and epithelial cell differentiation [1], and by a specific chromosomal translocation t(X:18)(p11:q11) [2,3,4]. The chromosomal translocation involves three variants of *SSX* (*SSX1*, *SSX2*, and *SSX4*) fused to the *SYT* gene (*SS18*) [5,6,7,8]. Mechanistically, the fusion protein SS18-SSX displaces wild-type SS18 protein from the BAF (Brg1-associated factor) complex, reversing the polycomb-mediated repression of *SOX2* gene transcription, leading to deregulated expression of SOX2—a transcription factor that regulates a large repertoire of genes critical in modulating cell growth and differentiation [9]. The three different fusion variants are similarly distributed in all three histologic subtypes, and were shown to be of some prognostic significance [10,11]. Both SYT-SSX1 and SYT-SSX2 have been shown to be equally potent in driving synovial sarcomagenesis in mouse models [12].

Synovial sarcoma occurs most frequently in the lower extremities, while it is less common in the upper extremities, head and neck and trunk [1,13]. For resectable early-stage disease, the main-stay of treatment approach is surgical resection, followed by adjuvant radiation with or without adjuvant chemotherapy [14]. In patients with locally-advanced tumor invading critical surrounding structures such as vessel or nerve, pre-operative radiation and/or chemotherapy may be used to downstage the disease first, followed by surgical resection. Some clinical features such as tumor size, margin status, histologic grade, age, sex, and bone and vascular invasion have been shown to be associated with outcomes, with larger tumor size (>5.0 cm) being consistently shown to be associated with shorter disease-free survival (DFS) and overall survival (OS) [11,15,16,17,18,19]. Some studies suggested that primary site may be associated with outcomes as well, but results were not consistent [15]. The benefit of adjuvant chemotherapy has been controversial in STS as a whole [14,20,21]. No synovial sarcoma-specific randomized clinical trial for adjuvant chemotherapy has been performed. Additionally, controversies remain regarding the benefit of metastatectomy in metastatic synovial sarcoma.

This is an observational study of patients with synovial sarcoma to investigate the clinical risk factors associated with stage IV disease at presentation and DFS, sarcoma-specific mortality, and OS in patients with early-stage synovial sarcoma.

## 2. Materials and Methods

Synovial sarcoma cases were identified from the Kaiser Permanente Northern California (KPNC) cancer registry between 1981 and 2014. The KPNC is a large integrated healthcare delivery system providing comprehensive medical care to more than 4 million members. The Kaiser Foundation Research Institute’s Institutional Review Board approved this study with waiver of consent (No. CN-14-2018-H), and funding was by a grant from the KPNC Community Benefit Program.

We identified a total of 154 cases that were histologically-confirmed synovial sarcoma between 1981 and 2014 through the electronic database of KPNC, and 130 of these cases were stage I to III. One case was excluded from the analysis due to a lack of relevant information in the medical chart. The diagnosis of all cases was confirmed by either typical chromosomal translocation and/or by expert sarcoma pathologists in academic institutions where the cases were submitted for consultation (including Stanford University, Mayo Clinic, Emory University, and Harvard Medical School). The majority of cases (approximately 90%) were confirmed by the typical gene rearrangement. The cytogenetic study was performed in the Regional Genetic Laboratory of KPNC that revealed the characteristic chromosomal translocation but did not identify the specific gene fusion variant [4]. The immunohistochemical staining was performed by the Regional Immunohistochemistry Laboratory of KPNC. Magnetic resonance imaging (MRI) was routinely performed for the initial assessment of the primary tumor and subsequent surveillance in most cases. Positron emission tomography/computed tomography (PET/CT) scan or CT scan of chest, abdomen, and pelvis was routinely performed for initial workup and surveillance every 3 to 6 months in the majority of cases.

Definition of primary site: proximal lower extremity includes gluteal, groin, thigh areas; distal lower extremity includes knee and below; proximal upper extremity includes shoulder, axilla, and arm; distal upper extremity includes elbow and below. Head and neck primary site includes maxillary sinus, infratemporal fossa, thyroid, floor of mouth, larynx, and neck; trunk includes axil, chest and abdominal wall, lung, pleura, mediastinum, pericardium, retroperitoneum, and infranephric area.

Statistical analysis: We compared demographic and clinical characteristics of the cases by early-stage and stage IV as well as by receipt of pre- and/or post-operative chemotherapy. All categorical variables were analyzed using the chi-square test and numerical variables were analyzed using the non-parametric Wilcoxon test and results presented with medians and interquartile ranges. In addition, Kaplan–Meier curves were created to display DFS and OS, and log-rank tests were used to compare differences between various strata such as early-stage versus stage IV disease and primary sites. Cox proportional hazard models were used to determine the impact of clinical characteristics such as tumor size, chemotherapy, and primary site on DFS, OS, and disease-specific survival after adjusting for age, gender, and race. All analyses were conducted using SAS version 9.3 (SAS Institute Inc., Cary, NC, USA).

## 3. Results

### 3.1. Immunohistochemical Staining Characteristics

To understand the pattern of immunohistochemical staining, we examined 43 cases of early-stage (35 cases) and stage IV (8 cases) synovial sarcoma diagnosed from 2006 to 2014. These cases had a more complete panel of immunostaining performed compared to the older cases. We presented the results of the most commonly performed markers in these cases (Table 1). BCL2 staining was positive in all 26 cases. Approximately 94% of cases stained positive for CD99 with a cytoplasmic and membranous staining pattern that was weaker and more diffuse than the typically homogeneous membranous staining seen in Ewing’s sarcoma. Pan-cytokeratin stained positive in approximately 80% of cases, while EMA 62%. Thirty-three percent stained positive for S100. Only one out of 33 cases (3%) stained positive for CD34. SMA and desmin stained negative in 100% of the cases stained. All 24 cases (100%) stained for Ki67 showed positive staining in a subset of tumor cells (ranging 5–50%).

### 3.2. Risk Factors Associated with Stage IV Disease at Presentation

The median length of follow-up for all cases was 7 years (range: 1 to 32 years). The age at the diagnosis ranged from 7 to 86 years old. Twenty-three cases (15%) presented with stage IV disease (metastatic disease) and with a median age of 50 (interquartile range (IQR) = 32)—significantly older than the median age of 36 (IQR = 25) for the patients with early-stage disease (stage I to III) (*p* = 0.02, Table 2). Sex and race/ethnicity distribution were not associated with stage of disease. For patients with stage IV disease, 70% presented with tumor size > 5.0 cm, compared to 53% for patients who presented with early-stage disease (*p* < 0.0001, Table 2). The median OS for patients with stage IV (2 years) was much shorter than that (25 years) of early-stage (Figure 1).

### 3.3. Risk Factors Associated with Poor Outcomes in Early-Stage Synovial Sarcoma

Next, we investigated the clinical features of the 130 early-stage patients and their association with DFS, sarcoma-specific mortality (sarcoma mortality), and all-cause mortality (mortality) (Table 3). Only four out of 130 cases were superficial disease. Eighteen cases (13.8%) underwent amputation, while 112 cases (86.2%) received limb-preservation surgery. Among the 112 cases who received limb-preservation surgery, eighty-one cases (72%) received either pre-operative or post-operative radiation, while thirty-one cases (28%) did not. The cases who did not receive radiation had a median tumor size of 3.0 cm, compared to 6.3 cm for the cases who received adjuvant radiation, indicating that smaller tumor size was the primary factor for not receiving radiation for the majority of patients.

Using Cox regression model and mutually controlling for demographic variables age, race, gender and clinical variables tumor size, whether chemotherapy was given or not, primary disease location, and histology in multivariate analysis, we found that tumor size > 5.0 was significantly associated with worse DFS (hazardous ratio (HR) = 2.9, *p* = 0.002), sarcoma-specific mortality (HR = 3.4, *p* = 0.003), and all-cause mortality (HR = 2.8, *p* = 0.003), compared to tumor size < 5.0 cm. In addition, compared to the patients with extremity and girdle synovial sarcoma, we found no difference in DFS for the patients with head and neck primary site as well as trunk primary site (Table 3). However, patients with head and neck and trunk primary site had significantly higher sarcoma-specific mortality (HR = 2.8, *p* = 0.04; HR = 3.2, *p* = 0.01) and all-cause mortality (HR = 2.5, *p* = 0.04; HR = 2.4, *p* = 0.03) (Table 3). No outcome difference was found between biphasic and monophasic histologic subtypes (Table 3). The median DFS and OS were 4.4, 3.6, and 2.6 years and 8.6, 7.9, and 3.3 years for extremity, head and neck, and trunk primary sites, respectively (Figure 2 and Figure 3).

### 3.4. Impact of Chemotherapy on Outcomes of Patients with Early-Stage Synovial Sarcoma

Among 130 patients with early-stage disease, forty received pre- and/or post-operative chemotherapy; among them, 30 (75%) had extremity or girdle primary. Eight (25%) of the 40 patients who received chemotherapy had amputation, while thirty-two (75%) had limb-preservation surgery, of which all received either pre-operative or adjuvant radiation. There was no statistically significant difference in age, sex, ethnicity, tumor size, primary site, or histologic type distribution between the patients who did and did not receive chemotherapy, though the patients who received chemotherapy had a higher percentage of T2 disease compared to the patients who did not receive chemotherapy (*p* = 0.07, Table 4). Twenty-two patients (55%) received pre-operative chemotherapy, 15 patients (37.5%) received post-operative chemotherapy, and 3 (7.5%) received both. The majority of patients (95%) received ifosfomide and adriamycin-based chemotherapy with a median number of four cycles (Table 5). Kaplan–Meier estimates showed no significant difference in DFS and OS between the patients who did and the patients who did not receive chemotherapy (Figure 4 and Figure 5). There was no difference in sarcoma-specific mortality (28% with no chemotherapy versus 35% with chemotherapy, *p* = 0.4) and all-cause mortality (35% with no chemotherapy versus 40% with chemotherapy, *p* = 0.5). Twenty-nine patients (72.5%) who received chemotherapy and 43 patients (52.2%) who did not receive chemotherapy had tumor size > 5.0 cm, and when the analysis was restricted to these patients, no significant difference in DFS and OS was found (data not shown).

### 3.5. Metastatectomy and Overall Survival of Patients with Lung Metastasis

Twenty-six patients had relapsed disease predominantly to the lungs after the initial treatment for early-stage synovial sarcoma; of these, 13 (50%) had metastatectomy. Most patients had multiple metastatectomy procedures performed. One patient had metastatectomy 8 times, three patients each had each 5, 4, and 3 such surgeries, and another three patients each had 2. One patient had 2 metastatectomies plus two radiofrequency ablation (RFA) procedures performed. Five patients had one metastatectomy only. Thirteen cases had disease in the lungs that was not resectable and did not receive metastatectomy. The OS for the patients treated with metastatectomy was 7.8 years, compared to 2.3 years for the patients not treated with metastatectomy, suggesting that metastatectomy may improve OS.

## 4. Discussion

Synovial sarcoma continues to be a very challenging disease to manage. The clinical workup often starts with biopsy of a suspicious mass that is typically round or oval with heterogeneous enhancement on imaging studies [22,23] followed by immunohistochemical staining of the biopsy specimen with a variable set of markers. All (100%) of the cases in our cohort stained were positive for BCL2, which is consistent with the previous studies [24,25]. BCL2 overexpression was the result of transcriptional activation rather than gene fusion or increased gene copy number [24]. More than 90% of our cases stained positive for CD99; however, unlike the strong and homogeneous membranous staining typically seen in Ewing’s sarcoma (or primitive neuroectodermal tumor), the CD99 staining in synovial sarcoma is usually weaker and more diffuse with both cytoplasmic and membranous distribution. CD99 is thought to inhibit neural differentiation, and may contribute to oncogenesis in Ewing’s sarcoma [26]. It is not clear if CD99 contributes to the oncogenesis of synovial sarcoma. Our data provide some helpful insight for the initial immunohistochemical evaluation of biopsy specimens that includes synovial sarcoma as a differential diagnosis.

The five-year survival rate of synovial sarcoma ranges from 37% to 90% in the literature [18,27,28,29]. Several clinical factors have been studied to characterize their impact on the prognosis. In a European non-randomized study of pediatric patients with synovial sarcoma, patients were grouped into low-, intermediate-, and high-risk based on tumor size and resectability. Twenty-six patients in the low-risk category with tumor size < 5.0 cm who were treated with surgery alone had 5-year event-free survival (EFS) at approximately 80% and OS at 90%, while 67 patients with high-risk features (larger unresectable tumor, axil primary, or with nodal involvement) had much worse EFS and OS [30]. Tumor size and primary tumor location have been shown to be significantly associated with DFS in several other studies [11,15,31,32,33]. Some suggested that distal or proximal location of extremity may be associated with difference in OS [15,17,33]. Our study did not find any outcome difference between distal and proximal extremity location (data not shown). While the correlative clinical characteristics in many studies are inconsistent, larger tumor size has been consistently shown to be the most significant prognostic factor secondary to metastasis at presentation. Consistent with this, our study with 130 cases of early-stage synovial sarcoma showed that tumor size > 5.0 cm was associated with approximately three-fold worse DFS, sarcoma-specific mortality, and all-cause mortality.

The 23 cases in our cohort who presented with stage IV had a poor OS of 1.3 years, similar to the previous studies [34,35]. The median age of patients presenting with stage IV disease was 50 years old—significantly older than the median age of the 130 cases who presented with early-stage disease (36 years old). The tumor size > 5.0 cm was significantly associated with the risk of presenting with stage IV disease. The trunk location (including axil primary site) appears to be associated with higher proportion of stage IV disease at presentation, but it was not statistically significant. We found that although there were no significant differences in DFS among extremity, head and neck, and trunk primary location, the head and neck primary and trunk primary were associated with approximately 2.5-fold higher sarcoma-specific mortality and all-cause mortality, compared to extremity and girdle primary site. This result appears consistent with the studies by Ferrari et al. [30,36].

The benefit of pre- and post-operative chemotherapy for STS has been an intense debate over the last two decades. There is no randomized clinical trial of pre- or post-operative chemotherapy specifically for synovial sarcoma. Only one randomized prospective trial, conducted by the Italian Sarcoma Group (ISG), showed OS advantage with adjuvant chemotherapy using epirubicin and ifosfomide in patients with extremity and girdle high-grade STS [37]. Other trials (mainly two large EORTC trials) have largely failed to demonstrate the benefit of adjuvant chemotherapy [38,39], though the meta-analyses have shown modest OS benefit [21,40,41]. The ISG followed-up with another randomized trial in pre-operative setting that tested three versus five cycles of chemotherapy, and showed that three cycles were not inferior [42]. A retrospective study of more than 300 cases of synovial sarcoma by Canter et al. with a pre-operative nomogram predicted an early survival benefit with ifosfomide and adriamycin-based chemotherapy [11]. Some other retrospective studies focusing specifically on synovial sarcoma also suggested survival benefit from adjuvant chemotherapy [29,43,44], though one study showed no benefit [33]. In our study, 40 patients with early-stage synovial sarcoma received pre- and/or post-operative chemotherapy, and showed no improvement in either DFS or OS. The higher percentage of T2 disease in the patients who received chemotherapy may or may not have influenced the outcomes. Since all 32 cases who had limb-preservation surgery and chemotherapy also received radiation, radiation would not be a factor influencing the benefit of chemotherapy. However, given its retrospective nature, the data can only be suggestive, not conclusive. A randomized chemotherapy trial specifically for synovial sarcoma in neoadjuvant or adjuvant setting would be of great significance.

The question about the benefit of metastatectomy remains inconclusive [35]. In our cohort, 13 of 26 patients who relapsed with predominantly lung metastasis received metastatectomy and had an OS of 7.8 years, compared to 2.3 years of patients who did not receive metastatectomy. In a previous study of 29 patients who received metastatectomy, five-year OS was 58.4% [45]. In a study of pediatric patients with synovial sarcoma (<22 years old), there was suggestion of improved OS with metastatectomy [33]. Improved OS with lung metastatectomy has been shown in other STS histologies [46,47]. However, the improved OS for patients who underwent metastatectomies could be due to their disease burden being more limited, rather than the procedure itself.

The limitations of our study include that it is a retrospective study with the data spanning more than 30 years, and the number of cases treated with chemotherapy was relatively small. The strengths of our study include the relatively large cohort of cases from a large integrated health care system, the long follow-up duration for most of the cases, and that the majority of patients who received chemotherapy were given a doxorubicin and ifosfomide-based regimen.

## 5. Conclusions

Our study showed that older age and large tumor size (>5.0 cm) are risk factors for presenting with stage IV disease and that tumor size > 5.0 cm, head and neck as well trunk primary are risk factors for poor survival in patients with early-stage disease. Future studies should continue to explore for the benefit of pre- and post-operative chemotherapy with newer agents.

## Figures and Tables

**Figure 1 medsci-06-00021-f001:**
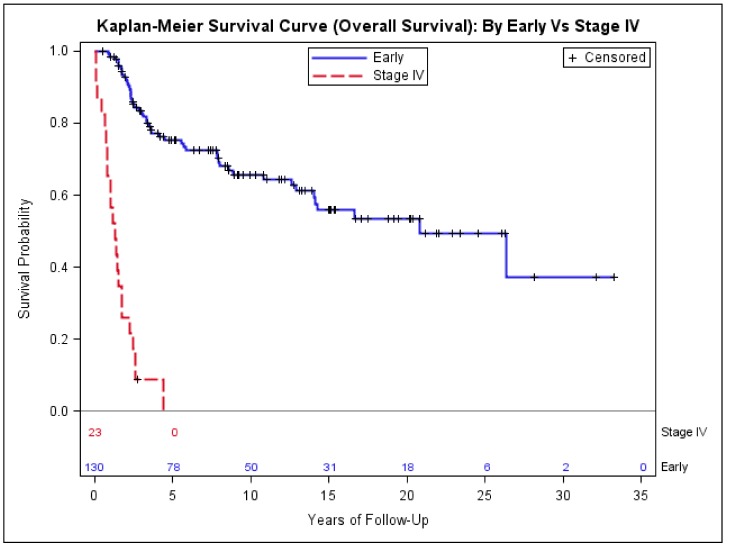
Kaplan–Meier estimates of overall survival of patients with stage IV and early-stage synovial sarcoma.

**Figure 2 medsci-06-00021-f002:**
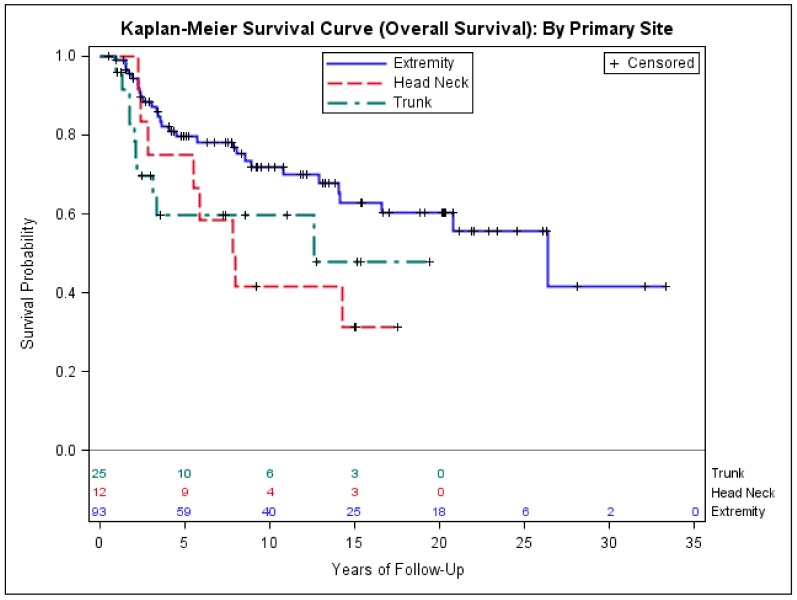
Kaplan–Meier estimates of overall survival of early-stage patients with extremity, head and neck, and trunk primary.

**Figure 3 medsci-06-00021-f003:**
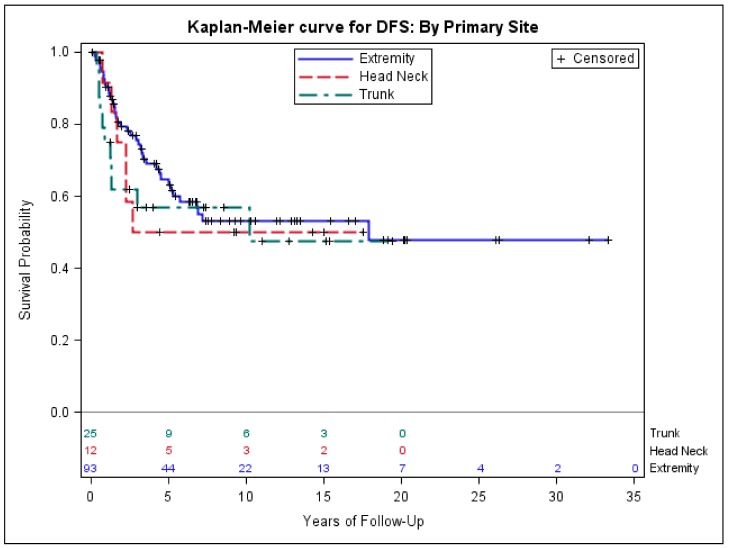
Kaplan–Meier estimates of disease-free survival of early-stage patients with extremity, head and neck, and trunk primary site.

**Figure 4 medsci-06-00021-f004:**
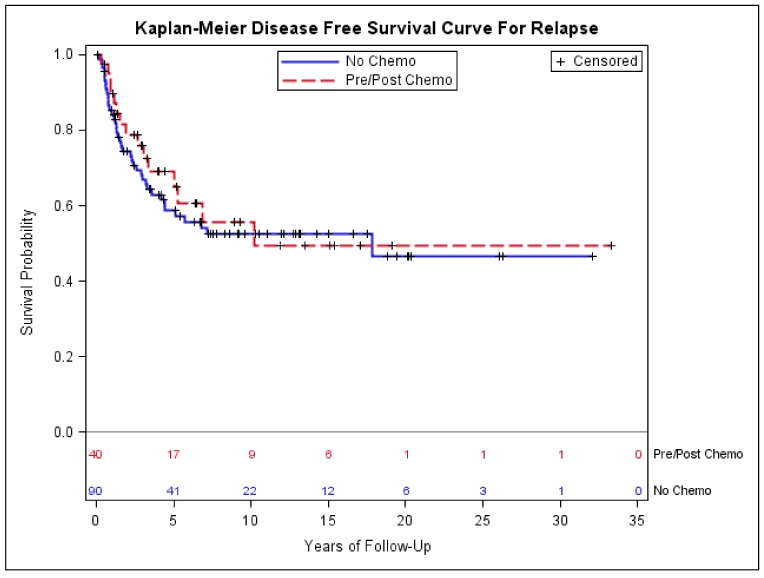
Kaplan–Meier estimates of disease-free survival of early-stage patients who did and did not receive chemotherapy.

**Figure 5 medsci-06-00021-f005:**
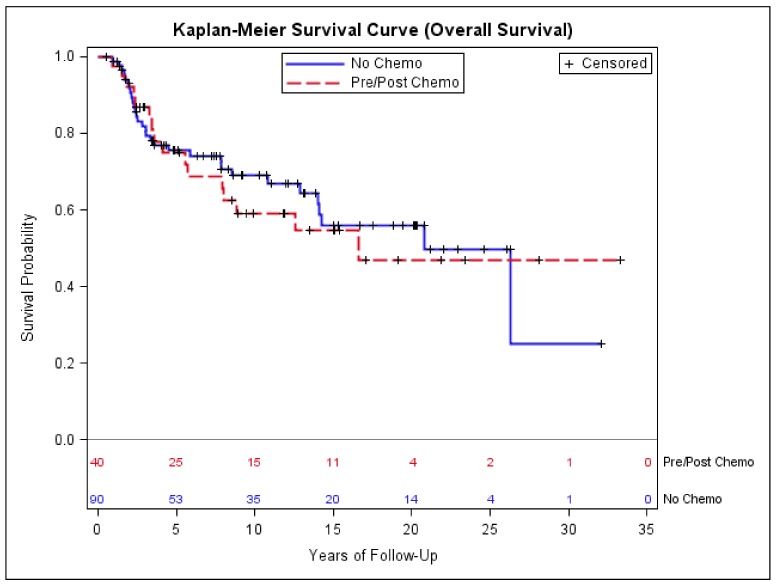
Kaplan–Meier estimates of overall survival of early-stage patients who did and did not receive chemotherapy.

**Table 1 medsci-06-00021-t001:** Immunostaining characteristics from 43 cases of early-stage and stage IV synovial sarcoma between 2006 and 2014.

	Positive	Negative	% Positive
BCL2 (*N* = 26)	26	0	100
CD99 (*N* = 32)	30	2	94
Pan-cytokeratin (*N* = 35)	28	7	80
EMA (*N* = 21)	13	8	62
S100 (*N* = 39)	13	26	33
CD34 (*N* = 33)	1	32	3
SMA (*N* = 38)	0	38	0
Desmin (*N* = 37)	0	37	0
Ki67 (*N* = 24)	24	0	100

**Table 2 medsci-06-00021-t002:** Characteristics of patients with stage IV and early-stage synovial sarcoma.

	Stage IV (*N* = 23 (%))	Early-Stage (*N* = 130 (%))	*p*-Value
Median age (IQR)	50 (32)	36.5 (25)	0.02 *
Sex			
Female	11 (48)	49 (38)	0.36 **
Race			
Asian	4 (17)	17 (13)	0.87 **
Black	1 (4)	11 (8)	
Hispanic	5 (22)	25 (19)	
White	13 (57)	77 (60)	
Tumor size (centimeter)			
<5.0	1 (4)	57 (44)	<0.0001 **
>5.0	16 (70)	69 (53)	
Unknown	6 (26)	4 (3)	
Primary Site			
EXTREMITY	13 (57)	93 (72)	0.10 **
HEAD_NECK	1 (4)	12 (9)	
TRUNK	9 (39)	25 (19)	
Histologic type			
Biphasic	5 (22)	50 (38)	0.18 **
Monophasic	13 (56)	67 (52)	
Poorly Differentiated	3 (13)	7 (5)	
Unknown	2 (9)	6 (5)	

IQR: interquartile range; * *p*-value determined using non-parametric Wilcoxon test; ** *p*-value determined using *Χ^2^* or Fisher’s exact test.

**Table 3 medsci-06-00021-t003:** Impact of tumor size, primary site, and histology on disease-free survival (DFS), sarcoma-specific mortality, and all-cause mortality for patients with early-stage synovial sarcoma.

Characteristic	DFS		Sarcoma Mortality		All-Cause Mortality	
	HR * (CI)	*p*-Value	HR * (CI)	*p*-Value	HR * (CI)	*p*-Value
Tumor size >5.0 cm vs. <5.0 cm	2.9 (1.5–5.5)	0.002	3.4 (1.5–7.5)	0.003	2.8 (1.4–5.8)	0.003
**Primary Site**						
Head and neck vs. extremity	1.2 (0.5–2.9)	0.759	2.8 (1.0–7.5)	0.049	2.5 (1.0–5.9)	0.04
Trunk vs. extremity	1.6 (0.8–3.3)	0.220	3.2 (1.3–7.7)	0.012	2.4 (1.1–5.5)	0.03
Histology: biphasic vs. monophasic	0.6 (0.3–1.2)	0.139	0.8 (0.4–1.5)	0.438	0.7 (0.4–1.3)	0.2

***** Hazardous ratio (HR) was adjusted for age, gender, race, and whether chemotherapy was given. CI: confidence interval.

**Table 4 medsci-06-00021-t004:** Characteristics of early-stage patients who did and did not receive chemotherapy.

	Chemotherapy Given (*N* = 40)	Chemotherapy not Given (*N* = 90)	*p*-Value
Median Age (IQR)	36 (29.5)	38.5 (25)	0.39
Sex (%)			
Female	12 (30.0)	37 (41.1)	0.23
Race (%)			
Asian	6 (15.0)	11 (12.2)	0.59
Black	3 (7.5)	8 (8.9)	
Hispanic	5 (12.5)	20 (22.2)	
White	26 (65.0)	51 (56.7)	
Tumor size (centimeter)			
<5.0	12 (30.0)	45 (50.0)	0.07
>=5.0	27 (67.5)	42 (46.7)	
Unknown	1 (2.5)	3 (3.3)	
Primary Site			
EXTREMITY	30 (75.0)	63 (70.0)	0.83
HEAD_NECK	3 (7.5)	9 (10.0)	
TRUNK	7 (17.5)	18 (20.0)	
Histology			
Biphasic	16 (40.0)	34 (37.8)	0.80
Monophasic	20 (50.0)	47 (52.5)	
Poorly Differentiated	3 (7.5)	4 (4.4)	
Unknown	1 (2.5)	5 (5.6)	

**Table 5 medsci-06-00021-t005:** Chemotherapy types and number of cycles received by patients with early-stage synovial sarcoma (A: adriamycin; I: ifosfomide; M: MESNA; V: vincristine; D: dactinomycin).

Type of Chemotherapy	Frequency (*N* = 40 (%))
AIM	30 (75)
Adriamycin/cyclophosphomide	5 (12.5)
Adriamycin only	1 (2.5)
MAID	1(2.5)
MAIV	1 (2.5)
VID	1 (2.5)
Dactinomycin/vincristine	1 (2.5)
**Number of Cycles**	**Frequency (*N* = 40 (%))**
1	1 (2.5)
2	4 (10)
3	14 (35.0)
4	12 (30.0)
5	3 (7.5)
6	6 (15)

AIM: adriamycin plus ifosfomide and MESNA; MAID: MESNA plus adriamycin and ifosfomide and dactinomycin; MAIV: MESNA plus adriamycin and ifosfomide and vincristine; VID: Vincristine plus ifosfomide and dactinomycin.

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
