# Peer review of "Risk Factors Including Age, Stage and Anatomic Location that Impact the Outcomes of Patients with Synovial Sarcoma"

_medsci, 2018, doi:10.3390/medsci6010021_

Reviewer 1 Report

Pan and Merchant present a study of risk factors associated with survival in patients with synovial sarcoma. While the analysis was conducted on a novel patient cohort, the results do not dramatically shift the paradigm on how we think of treatment care for synovial sarcoma. With that said, the data is important for the medical community to have, and there are some finer points that could be emphasized a little more in the introduction and discussion.  As written, the introduction concludes with the short paragraph about what the study entails, but the study offers more and should be expanded upon especially in regard to the controversial use of chemotherapy and the surprising benefits seen in patients who underwent metastasectomy at their institution.

The science was logically presented and sound in the approach and analysis. With some grammatical corrections and a small shift to highlight the points mentioned above, this could be a better manuscript. Below is a list of more specific recommendations to improve the manuscript.

Response: We appreciate the reviewer’s positive feedback and agree that our manuscript adds important value to the sarcoma knowledge and research.

1)      Include citation -already there several papers on the topic and one from 1996 (not cited) that looked at similar prognostic factors and concluded similarly that larger tumors (> 5 cm) had an impact on survival (PMID: 8648375).

Response: we have added this reference.

2)      At least once in the methods section, describe that the early stage synovial sarcomas includes stages 1-3. It is mentioned later in the paper, but it should be defined earlier.

Response: we have added it to the Method section.

3)      Also please stratify how many synovial sarcoma cases were confirmed with FISH translocation data versus trained pathological identification.

Response: We have added the data to the Method section.

4)      In the discussion, (pg 10) it states that BCL2 overexpression was the result of transcriptional activation rather than gene fusion or increased copy number. This implies that these tumors were genomically sequenced and/or transcriptionally sequenced, which if that is the case the data should be highlighted. More details are needed on how the authors arrived at this conclusion.

Response: BCL2 expression was detected by IHC, not by sequencing.

5)      Also on pg 11, the authors’ data did not provide any insights into the immunohistochemical evaluation of synovial sarcoma as all markers tested in the study are standard makers used in clinical diagnoses.  Please rephrase this sentence. 

Response: We have made this revision.

Reviewer 2 Report

I enjoyed reading this manuscript. However I have a number of concerns.

1)  The time frame covered is very long: 1981 – 2014. During this time course the diagnosis of synovial sarcoma has improved and refined considerably. Therefore, are the authors sure that all cases are in fact synovial sarcoma? Have these cases been re-reviewed and had molecular confirmation?

Response: Yes, in fact, all the cases were re-reviewed by an academic expert and confirmed by the typical gene translocation.

2)  Furthermore, the treatment algorithms during these 33 years would have varied thus impacting the interpretation of the findings.

Response: The treatment has changed little during the past 3 decades, unfortunately, especially for early stage disease.

3)  No comment can really made regarding neoadjuvant/ adjuvant chemotherapy from a retrospective study.

Response: We agree with this.

4)  This study adds little to our current knowledge of synovial sarcoma.

Response: We believe our manuscript adds good value to the current knowledge, including the risk factors associated with prognosis that are still not clear and our metastatectomy data though small confirms the benefit of such procedure.

5)   Size is a continuous variable and although arbitrary cut offs are often used, it is known that small synovial sarcomas can result in metastatic disease.

Response: We agree with this. Size<5.0 cm="" still="" has="" risk="" of="" though="" lower="" than="" size="">5.0 cm.

Reviewer 3 Report

Dear Editor:

Thank you for the opportunity to review the paper by Pan et al. entitled “Risk factors that impact the outcomes of patient with synovial sarcoma”.  This is a retrospective review of synovial sarcoma patients treated at Kaiser Permanente.  Given the volume of sarcoma patients see at Kaiser in California, publication of its data sets is strongly encouraged.  Before final acceptance, there are some issue that should be addressed as listed below.

 Response: Thank you for your very positive comment.

1)  The average age of synovial sarcoma patients are between 15-40. (Stacchiotti S, Van Tine BA: Synovial Sarcoma: Current Concepts and Future Perspectives. J Clin Oncol 36:180-187, 2018). The average age of both metastatic and (36.5) and non-metastatic (50) patients in this manuscript is greatly shifted towards older patients.  The manuscript, including the title, should reflect this.  It is interesting, from an adult management perspective, to have papers on older patients.

Response: We have changed the title as suggested. 

2)  The authors state that chemotherapy has no effect on their outcomes.  The numbers are too few and the regiments are too many to make any strong conculsions.  The article should be toned down from this presepctive.  Also they should look at the dose of ifosfamide that was given (at it is dose dependent) for their analysis.

 Response: We agree with this comment and have made the change (page 14 in Discussion section).

3)  The KM curvers for section 3.5 should be shown and the number of lesions involved in the metastateecomy should be analyzed to determine if it was all signle lesions that were removed or many.

Response: We have added additional information to the Discussion section at page 10.

Reviewer 4 Report

Author describes their experience of risk factors in adults (or children, we don’t know?) synovial sarcoma. In my opinion, this paper is poorly written, associate pathologic findings, risk factors analysis and so many nonscientific conclusions that avoid this paper to be published per se in a Journal.     

Response: We disagreed with this comment.  All other reviewers have commented that our manuscript is well written.                 

General presentation:

I am surprise that figures/Tables are included in the text and not at the end of the manuscript.

Response: They are at the end of the manuscript, but formatted by the journal to be weaved in the text for the convenience of the reviewers.

Major comments:

Abstract: it may be interesting to have the total number of analyzed cases here. As authors talk about pediatric and adults’ tumor in the background, we guess that this analyzes concerns all patients age. It that correct? I prefer to consider that OS is a % and “survival time” is a delay, but I think that an OS of 1.3 y is not a correct way to write this, may be it is a median overall survival time/delay? Is there a difference between “disease specific survival” and “sarcoma specific mortality”? If not, please consider to us the same terms. Please consider that “patients” do not relapse. But only their tumors relapse. So please correct this in the entire document. Please add numbers in the results text as much as possible.”3 subtypes”: are you talking about FNCLCC grade I vs. II vs. III? Conclusions should not repeat the information already written in the results but give some additional findings.

Introduction: line 51, but many other studies showed no impact of the fusion transcript”, so please balance this assertion,

Response: We show no impact with the fusion transcript too in our manuscript.

Inclusion criteria: age? Pathology definition of SS in 1981 was probably different than in 2014. Could you comment on this? What were the therapeutic guidelines during this period?

Response: We include all cases with synovial sarcoma. There have been little difference between then and now, except the discovery of the fusion genes.

Could you explain if immune-staining has been done for this paper or was it a review of the pathologic reports included in the database? Please add this information in the “material and method” chapter and in the objectives of this study.

Response: We have added this information.

Table 2: what about tumor at 5 cm? Color skin is not a “race”; Asia is considered a continent.

Response: there was no tumor that was exactly 5 cm in our cohort.

Figure 1: please add the OS of the 2 arms.

Response: OS was 25 years for early stage, and 2 years for stage IV.

Median age of you entire cohort?

Response: we have added this data to the result section.

Line 137, definition of stages?

Response: Stage was defined by AJCC staging system.

Line 154, definition of “superficial disease”?

Response: as defined by AJCC staging system, tumor that did not invade musculature.

Line 152, I am surprise that all early stage cases concern only limbs primary. What about stage I-III trunk wall primary?

Response: we included trunk cases as well.

Line 159: it is not because you found a difference in size that this was the reason why the physician decided this. May be this was due to the primary or the age of the patient or another factor.

Response: We agreed with this.

Line 163, you analyze the role of chemotherapy, but we do not know how many patients receive it, when, why and how.

Response: this data is clearly in our manuscript.

Line 172, only 2 subtypes analyzed here (and 3 in the abstract …). Why?

Response: because there were just 7 cases with poor differentiated subtype, not sufficient to be included into analysis.

Why didn’t you analyze primary role in 3 groups? Head and neck vs. trunk vs. limbs? It may be easier to understand. Please explain why you show primary results in Table 3 and not the other risk factors analyzed in the multivariate analyze.

Response: we did analyze the data with multivariate analysis

Line 194: no other primary than limbs here again because you only speak about amputation or limbs salvage surgery?

Line 228: this comparison is really scientifically strange without any analyzes of other risk factors? We don’t know anything about all tumor characteristics and authors only analyze only one finding (metastasectomy). This analyze should only be include in a large multivariate analyze to consider if this is a bias or not. For instance, may be pulmonary relapses are more numerous in case of limbs primary and that is the reason why lung resection is less feasible in such situation and why survival looks better. In my opinion, this chapter should be cancelled.

Response: we disagreed with this comment. Other reviewers found value with this data.

Minor comments:

Table 4: UNKN? IQR?

Table 5: correct, ifosfamide

Response: it is ifosfomide.

Ref: The large pediatric SIOP European MMT study is not referenced here [Orbach D 2011]

Response: This article concerned mainly about radiation for pediatric patients with synovial sarcoma, not relevant to our manuscript. Please see below part of abstract from the article: The aim of this analysis was to identify if the modified indications of radiotherapy (RT) or radical surgery compromised survival in pediatric synovial sarcoma (SS).

Round  2

Reviewer 2 Report

1) The number and percentage should be used throughout the manuscript.

Response: Done

2) Further to my previous review, the authors should discuss the limitations of the study and the challenges of the available evidence base. 

Response: Done